# Defect Detection Algorithm for Wing Skin with Stiffener Based on Phased-Array Ultrasonic Imaging

**DOI:** 10.3390/s23135788

**Published:** 2023-06-21

**Authors:** Chuangui Wu, GuiLi Xu, Yimeng Shan, Xin Fan, Xiaohui Zhang, Yaxing Liu

**Affiliations:** 1State-Owned Machinery Factory in Wuhu, Wuhu 241007, China; wcg_994@163.com (C.W.);; 2College of Automation Engineering, Nanjing University of Aeronautics and Astronautics, Nanjing 211106, China; 3Anhui Province Aviation Equipment Testing and Control and Reverse Engineering Laboratory, Wuhu 241007, China

**Keywords:** wing skin with stiffener, defect defection, phased-array ultrasonic imaging, data post-processing, real-time imaging

## Abstract

In response to the real-time imaging detection requirements of structural defects in the R region of rib-stiffened wing skin, a defect detection algorithm based on phased-array ultrasonic imaging for wing skin with stiffener is proposed. We select the full-matrix–full-focusing algorithm with the best imaging quality as the prototype for the required detection algorithm. To address the problem of poor real-time performance of the algorithm, a sparsity-based full-focusing algorithm with symmetry redundancy imaging mode is proposed. To address noise artifacts, an adaptive beamforming method and an equal-acoustic-path echo dynamic removal scheme are proposed to adaptively suppress noise artifacts. Finally, within 0.5 s of imaging time, the algorithm achieves a detection sensitivity of 1 mm and a resolution of 0.5 mm within a single-frame imaging range of 30 mm × 30 mm. The defect detection algorithm proposed in this paper combines phased-array ultrasonic technology and post-processing imaging technology to improve the real-time performance and noise artifact suppression of ultrasound imaging algorithms based on engineering applications. Compared with traditional single-element ultrasonic detection technology, phased-array detection technology based on post-processing algorithms has better defect detection and imaging characterization performance and is suitable for R-region structural detection scenarios.

## 1. Introduction

As an important component of an aircraft, the wing primarily functions to provide lift and, together with the tail, ensure the aircraft’s stability [1,2,3]. Regular inspection of wing structures to maintain an aircraft’s good condition is an important guarantee of the combat effectiveness of military aviation [4,5]. The R region of the wing is an area of concentrated stress that is prone to fatigue damage due to long-term force during flight, leading to the appearance of internal defects, such as cracks and voids, and posing a significant safety hazard during the operation of the wing [6,7,8,9]. Accurate non-destructive testing of structural defects in the R region of wing skin with ribs is of great practical significance.

Non-destructive testing is a technique that detects the surface or internal structure parameters and organizational status of a test sample by utilizing changes in thermal, acoustic, optical, electrical, magnetic, or other reactions caused by the presence of anomalies or defects in the material structure, without damaging the material performance and structural integrity [10]. Ultrasonic testing is a well-established and widely applied non-destructive testing method. Compared with other non-destructive testing techniques as shown in Table 1, ultrasonic non-destructive testing has a wide detection range, large detection depth, high output sensitivity, and accurate defect location [11,12]. Furthermore, due to its low cost, reliable and stable performance, and harmlessness to humans, ultrasonic non-destructive testing has gradually become a promising technique for aviation industry applications at home and abroad [13].

Ultrasonic phased-array technology is a major research direction in the field of ultrasonic testing. A phased array is composed of multiple piezoelectric crystal elements that are arranged in a regular pattern and can be excited and controlled by an electronic system to achieve control and adjustment of beam focusing position and deflection direction [14,15]. Due to the ultrasonic phased array’s strong defect detection capabilities and the presence of intricate structural defects in the wing skin’s R region with ribs, employing an ultrasonic phased array for defect detection in the R region of wing skin has emerged as a practical and viable solution.

Compared to traditional single-element ultrasound detection technology, ultrasound phased-array inspection can achieve beam focusing in the imaging area, resulting in higher-quality defect echoes and a higher signal-to-noise ratio in the image [16]. Phased-array ultrasonic testing systems have a wider scanning range and are more adaptable to complex detection scenarios than traditional ultrasonic testing systems. Compared to existing ultrasound detection technologies, ultrasound signal post-processing can flexibly process the acquired ultrasound signals, thereby obtaining better defect reconstruction images [17,18]. Applying data post-processing techniques to ultrasound phased-array inspection systems can not only fully leverage the electronic control advantages of phased-array systems but also effectively improve imaging quality and defect detection rate by combining them with the efficient digital processing performance of computer systems [19,20].

It should be noted that the mainstream ultrasound data post-processing technologies currently mainly focus on known defects, while the complexity of defect situations in practical engineering and unknown information make defects prone to false alarms and missed detections, leading to a decrease in algorithm applicability and unsatisfactory imaging results [21,22]. Based on the above issues, it is necessary to optimize and improve defect-imaging algorithms for actual inspection structures.

This study proposes a phased-array measurement and imaging scheme for detecting defects in reinforced wall panels in the R region, models and analyzes the post-processing imaging algorithms for ultrasonic detection, and evaluates the best focusing imaging method for defect detection characterization. Meanwhile, improvements to real-time imaging are proposed based on the reciprocity of sound fields and sparse theory. Imaging noise artifacts are suppressed through adaptive beam synthesis and equal-echo-path pseudo-image removal methods.

## 2. Analysis of Ultrasonic Testing Imaging Scheme

As shown in Figure 1, based on the actual structure of the reinforced wall panel R area to be inspected, a linear ultrasonic phased-array transducer was selected as the ultrasonic measuring device, and a reference coordinate system was established for the imaging area to determine the detection and imaging scheme of the phased-array back-to-back scanning. Meanwhile, according to the principles of ultrasonic signal post-processing characterization, mathematical models were developed for three mainstream data post-processing algorithms: coherent compound plane-wave imaging, multi-element synthetic-aperture imaging, and full-matrix–full-focusing imaging. Based on the simulation results of beamforming imaging, the detection sensitivity and resolution performance of different algorithms were qualitatively analyzed, and the imaging quality of the algorithms was quantitatively evaluated using the −6 dB API index.

Through acoustic-field analysis simulation software Matlab FieldⅡ, based on the theoretical foundation of linear acoustics, multiple-beam focusing and amplitude variation can be set to simulate and analyze the acoustic field and echo signal of the ultrasonic transducer excitation process, with high computational accuracy. The specific imaging simulation process is divided into the following four parts:(1)Firstly, the acoustic-field simulation environment is initialized, and based on the experimental results of the calibrated measurement of the detection specimen and the actual indicators of the ultrasonic phased-array probe, the acoustic imaging simulation environment parameters and probe array parameters are set, with specific input parameters shown in Table 2.(2)As shown in Figure 2, the linear phased-array probe was modeled based on the input parameters, and the impulse response of the transmission and reception apertures was set to a sinusoidal wave. The excitation signal was a Hanning windowed, 3-cycle, 5 MHz sine wave, and the excitation signal waveform and impulse response waveform are shown in Figure 3a.(3)In order to evaluate the imaging characterization performance of different data post-processing algorithms, a group of point scatter defects with different depth distributions were set in the planar space, and the algorithm was evaluated by simulating the imaging of the reference defects. The spatial positions of the 5 point defects are shown in Figure 3b, with a characterization range of a spatial plane of 50 mm × 50 mm, an imaging range of 5 mm–55 mm along the *z*-axis, an imaging range of −25 mm–25 mm along the *x*-axis, and an imaging grid spacing of 0.1 mm.(4)Through linear simulation, the sound field distribution and reflection defect echo data of the phased array are calculated. Based on the imaging algorithm, the phased-array echo data is subjected to time-delayed stacking, and the focal point intensity values within the imaging area are computed to obtain the ultrasound detection imaging results.

The following are the simulation results of three post-processing algorithms, namely coherent plane-wave compounding imaging (CPWC) (Figure 4), multi-element synthetic-aperture imaging (MSAF) (Figure 5 and Figure 6), and full-matrix/full-focusing imaging (FMC-TFM) (Figure 7).

To quantitatively analyze the lateral resolution of post-processing imaging algorithms, an array performance parameter (API) is introduced to evaluate the imaging performance of the array for point-like scatter defects as shown in Figure 8. A smaller API indicates higher system resolution. Taking the −6 dB attenuation of the maximum beam amplitude as the reference threshold, the API value is defined as the ratio of the imaging area A −6 dB where the composite beam amplitude is higher than the threshold to the ultrasound wavelength λ^2^.

The API parameter, which characterizes five scattering points within the characterization range, is calculated as the average output for evaluating detection resolution. The API parameter is used to quantitatively analyze the resolution of the coherent composite plane-wave imaging, multi-element synthetic-aperture imaging, and full-matrix–full-focusing imaging algorithms, and the calculation results are shown in Table 3.

Based on the quantitative analysis of the −6 dB API parameters, the following main conclusions can be drawn: (1) the full-matrix–full-focusing (FMC-TFM) imaging algorithm has the best imaging performance, with high lateral resolution and almost no artifacts, providing clear contrast for defect imaging. (2) The multi-element synthetic-aperture (MSFA) imaging algorithm produces good imaging quality when the sub-aperture size M = 4, with a certain degree of lateral resolution for defect imaging, but with relatively more artifacts and lower signal-to-noise ratio. (3) The hybrid plane-wave (CPWC) imaging algorithm has poor imaging quality and the lowest lateral resolution, with a certain suppression effect on the ultrasonic grating and fewer artifacts. Based on simulation analysis, the FMC-TFM algorithm has the best overall imaging performance. Considering the complex detection structure of the R-region components, the interference of stray noise and imaging artifacts during actual detection, and the high demand for detection system stability in field testing environments, the FMC-TFM algorithm is recommended as the imaging algorithm for detection, and subsequent improvements should be made according to actual needs.

## 3. Improvement of Ultrasonic Imaging Algorithms

As shown in Figure 9, the FMC-TFM imaging is based on full transmit–receive array element virtual focusing for imaging, which has excellent imaging characteristics [12]. However, the collection and delay calculation of echoes for full-matrix data significantly reduce imaging efficiency. As the number of elements increases, the imaging time exponentially increases, making it impossible to meet the real-time inspection requirements in engineering. Therefore, improvements are needed to enhance the real-time imaging capability of the FMC-TFM algorithm.

To address the long computation time of the FMC-TFM algorithm, a semi-matrix acceleration algorithm for full-focusing imaging is proposed based on the reciprocity theorem of the phased-array echo data, which reduces the redundancy of matrix data as shown in Figure 10. Additionally, the sparse imaging mode of the FMC-TFM algorithm is proposed by implementing sparse elements, sparse transmission, and full-aperture reception, based on the theory of sparse data sampling, to achieve a sparse approximation of full-matrix data. Furthermore, through simulation and comparative evaluation of multi-element full-focusing algorithms, the imaging performance of the sparse full-focusing mode is validated. Finally, combining the semi-matrix and sparse-matrix acceleration modes as shown in Figure 11 and Figure 12, a simplified FMC-TFM algorithm is proposed using a GPU matrix parallel computing framework to achieve the most efficient imaging for full focus in the space domain. The schematic diagram is presented in Figure 13 and simulation results are presented in Figure 14, Figure 15, Figure 16 and Figure 17.

In response to the problem of low imaging speed and inability to meet imaging scanning requirements of the full-matrix–full-aperture algorithm, based on the analysis of acoustic reciprocity and sparse sampling theory, we proposed a de-symmetry redundant imaging mode for the sparse full-aperture algorithm through multi-element full-aperture algorithm simulation, comparison, and evaluation. Meanwhile, by analyzing the processing of imaging-point operations, we eliminated the calculation redundancy in the single-frame imaging process, separated the channel data delay calculation and the echo signal superposition process, and proposed the simplest full-aperture algorithm. Combined with a GPU parallel processing framework, the imaging speed was greatly accelerated while ensuring imaging quality, which can meet the engineering scanning requirements. A single-frame image takes only about 0.5 s. Finally, through qualitative and quantitative analysis of simulation experiments, the imaging performance of the simplest full-aperture algorithm was validated.

## 4. Improvement of Noise Artifacts in Ultrasound Imaging Algorithms

This chapter focuses on the problem of noise artifacts that affect the quality of imaging in full-focusing ultrasound imaging. By mathematically modeling and analyzing the minimum-variance non-destructive response and coherent-factor adaptive beamforming methods, along with also evaluating the simulated imaging results of various defect types, adaptive filtering and improvement of noise artifacts in ultrasound imaging are achieved, and high-quality and robust improvements to the full-focusing algorithm are realized.

Traditional beamforming methods have a simple and reliable imaging principle, involve low computational complexity, and can easily achieve real-time imaging in hardware. However, the simple delay-and-sum beamforming method has limited suppression effect on noise artifacts and is prone to interference that covers the desired signal in actual detection environments, affecting the resolution and contrast of ultrasound imaging. Although amplitude variation can improve imaging contrast to some extent, it cannot be universally applicable to all detection situations, and it will reduce the image resolution. Adaptive beamforming dynamically determines the aperture weights based on the echo feature information, enhances the effective signals and suppresses the clutter signals separately, and can effectively separate off-axis scattering and noise from the desired signal, improve the directional transmission of the transmitted signal, and suppress the noise artifacts in ultrasound images.

### 4.1. Adaptive Beamforming

Adaptive beamforming is a beamforming algorithm that adaptively adjusts the signal superposition based on the characteristics of the echo signal. It dynamically determines the aperture weights based on the characteristics of the echo signal and enhances and suppresses the effective and clutter signals separately [13]. This algorithm can effectively separate off-axis scattering and noise from the desired signal and improve the directionality of the transmission signal, thus suppressing noise artifacts in ultrasound images [14].

Adaptive beamforming can be mainly divided into two categories: minimum-variance distortionless response (MVDR) and coherent-factor (CF) beamforming. The MVDR beamforming algorithm optimizes the weighted superposition of the aperture emission-reception channel data, minimizing the beamforming variance while keeping the target signal power constant [15]. The CF beamforming algorithm calculates the coherence of all channel signals at the imaging point and uses it as the imaging weight to weight the output of signal delay and superposition, achieving off-axis signal suppression. The two types of adaptive beamforming methods have their respective applicability under different imaging detection conditions [16].

Adaptive beamforming methods are often used in synthetic-aperture imaging algorithms. In order to address the problem of noise artifacts affecting imaging quality in full-focusing ultrasound imaging, the effective transfer of adaptive beamforming algorithms is achieved by combining the imaging requirements of R-region component detection with the MVDR and CF beamforming algorithms to achieve adaptive improvement of imaging noise artifacts.

### 4.2. Simulation and Evaluation of Defect Imaging

The adaptive beamforming algorithm dynamically determines the aperture weights based on the echo signal characteristics, which can effectively suppress off-axis signals and improve imaging noise artifacts. Through mathematical modeling and analysis, the implementation principles and computation processes of the minimum-variance distortionless response (MVDR) and coherent-factor (CF) adaptive beamforming methods have been studied, and the application of beamforming based on full-aperture imaging has been improved. In order to verify the effectiveness of the adaptive beamforming method in improving imaging, multiple defects were simulated and the DAS, MV, ESBMV, CF, GCF, and SCF beamforming algorithms were compared and analyzed by image reconstruction. (DAS: Digital subtraction angiography. MV: Minimum variance. ESBMV: Eigenspace minimum-variance beamformer. CF: Chlorophyll fluorescence imaging. GCF: Generalized coherence coefficient. SCF: Self-consistent field method.)

The specific parameters were set as follows: a 64-element linear array probe with a center frequency of 5 MHz and an element spacing of 0.5 mm, a system sampling rate of 100 MHz, and a 3-cycle length sinusoidal excitation signal with a Hann window (Figure 18).

The imaging mode of the simplest full-focusing algorithm was used to control the element reception, and the imaging area of the set point defects was simulated and the echo signals were collected according to the simulation process. Combining with the image reconstruction process of the simplest full-focusing algorithm, the adaptive methods of MV class and CF class were used to compute and synthesize the ultrasonic beam, and the two-dimensional defect morphology of the imaging area was characterized by image reconstruction. This is shown in Figure 19.

As shown in the figure, under the interference of global Gaussian noise, the imaging region of the DAS beamforming algorithm is completely submerged in high-energy noise, which seriously affects the characterization and identification of the defect points, especially in the far-field area of ultrasonic detection. As the energy of the detection signal decays, the imaging characterization of the defects is almost invisible in the noise signal. The imaging results of the CF beamforming algorithm can effectively improve the imaging characterization of defects under noise interference, but due to the excessive suppression of incoherent signals, the defect information at the bottom of the imaging region cannot be characterized. The defect signals at points 4 (depth 29 mm) and 5 (depth 37 mm) are mistakenly filtered out as noise. The imaging of GCF beamforming also suffers from the same problem, and due to the excessive suppression of noise signals, the adaptive imaging characterization of the bottom point defects is impossible. However, compared with CF beamforming, it can display the imaging of defect point 4. In contrast, the SCF, MV, and ESBMV beamforming algorithms can output the imaging of all point defect information while suppressing noise interference and improving imaging characterization performance. Among the three, the SCF algorithm has stronger global noise suppression performance, while the MV and ESBMV algorithms can better improve the imaging distortion of point defect 1 (depth 5 mm) in the near-field region.

To comprehensively evaluate the adaptive beamforming algorithms, the ultrasonic imaging simulation process using the simplest full-focusing algorithm is also adopted to detect and image a set of horizontal and a set of oblique linear defects, separately. The two-dimensional defect characterization results of the imaging regions are shown in Figure 20 and Figure 21.

Based on analysis, the SCF beamforming algorithm shows the optimal defect-imaging performance for both point and line defects. Considering the random distribution of discrete interference in practical engineering inspections, the application of the SCF beamforming algorithm in defect detection of the rib-stiffened wall panel R zone of aircraft wings can significantly suppress off-axis signals and noise interference, thus improving the robustness and stability of the ultrasound detection imaging system.

### 4.3. Equivalent-Path-Length Pseudo-Echoes Dynamic Elimination

Isochronous pseudo-echoes are erroneous ultrasound local images caused by the non-uniqueness of the time delay index of imaging points in the emission-reception channel data [17]. Considering that the echo amplitude of the target body attenuates less and is stronger when it is closer to the surface, the phenomenon of near-surface pseudo-echoes will be more severe. At the same time, if there is acoustic noise or a trailing phenomenon in the echo signal, the isochronous pseudo-echoes will continue to worsen and even flood a large amount of effective information in the acoustic field [18]. In the imaging area of the wing-spar wall plate that needs ultrasonic detection, the distance between the imaging area and the phased-array transducer is short, the structure of the near-surface area of the component in the R region is complex, and there is a lot of off-axis interference in the acoustic field, making it more prone to the problem of isochronous pseudo-echoes [19].

In the ultrasonic full-focusing algorithm, each imaging point is virtually focused by time-delayed superposition of emission-reception channel data, and effective matrix data is traversed for acoustic-field intensity reconstruction imaging [20]. Through the analysis of the acoustic-field distribution in the imaging area, it can be known that each channel datum corresponds to an isochronous line with an equal sound intensity distribution for each imaging point. The isochronous line with an intensity higher than the threshold is called the effective isochronous line [21,22]. If the number of times the effective isochronous line is superimposed on the imaging point is less than a certain proportion of the total superimposed times, the point is judged to be a non-defective point. Based on the root-mean-square-error method and the effective channel data superimposed output mode of the simplest full-focusing algorithm, a bilateral root-mean-square calculation method (B-RMSM) for the threshold is proposed:


(1)The envelope $x_{e}(t)$ of the echo signals collected from E effective channels is obtained by Hilbert transform:(1)$$x_{e}^{\prime }(t)=x_{e}(t)*\frac{1}{\pi t}=\frac{1}{\pi }\int_{-\infty }^{+\infty }\frac{x_{e}(t)}{t-\tau }d\tau$$(2)The minimum peak value $I_{e}^{\prime }$ of the E echo envelopes is determined, and the mean value Mean ($I_{e}^{\prime }$) is calculated:(2)$$\mathrm{Mean}\left(I_{e}^{\prime }\right)=\sum_{e=1}^{E}I_{e}^{\prime }/E$$(3)The mean value Mean ($I_{e}^{\prime }$) is selected as the binary threshold, and the peak values $I_{e}^{\prime }$ of small envelopes are divided into two sets, C1 and C2.(4)The variances σ_1_ and σ_2_ of $S_{i}$ data points in sets C1 and C2 are calculated, respectively:(3)$$\sigma_{i}=\sqrt{\frac{1}{S_{i}}\sum_{n=1}^{S_{i}}{\left(C_{i}\left(n\right)-\bar{C_{i}}\right)}^{2}},i=1,2$$(5)The threshold value Δ*I* of the effective is defined as follows:(4)$$\Delta I=\sqrt{\mathrm{Mean}{\left(I_{ij}^{\prime }\right)}^{2}-\frac{\left(\sigma_{1}+\sigma_{2}\right)}{4}}$$



When the sound field intensity of an isochronous line exceeds a threshold, the isochronous line is considered as an effective isochronous line. Based on the threshold of effective isochronous lines, an algorithm for isochronous error suppression (EAEM) is proposed. Each valid data channel of the imaging point is discriminated sequentially, and the proportion of effective isochronous lines is counted. If the number of effective isochronous lines for the imaging point exceeds 1/s of the total, the point is considered a true defect point, and the superimposed amplitude of the imaging is preserved by adaptive weighting using the CF method. Otherwise, the point is an isochronous error point, and the CF-weighted result of the minimum value in the echo data channel for the point is used as the sound field intensity for imaging. Here, s is a dynamic critical parameter. The overall calculation process of the adaptive suppression algorithm for isochronous error is shown in Figure 22.

To validate the effectiveness of the equal-path-length artifact removal algorithm, a linear defect distribution was set up in the near-field area of the imaging region, and the simulated defects were reconstructed and characterized based on the simulation imaging process of ultrasonic testing. Combined with the simplest total-focusing method and the self-adaptive coherent-factor beamforming algorithm, the imaging results with and without the equal-path-length artifact removal algorithm were compared to analyze the effectiveness of the artifact removal algorithm and the improvement in imaging performance. The simulation results are shown in Figure 23.

Through the above simulation experiments, the effectiveness of the equal-sound-path pseudo-image removal algorithm has been verified, which can improve the imaging characterization quality of the full-focusing algorithm in the ultrasonic detection of defects in the R region of the rib-stiffened wall panel.

In this paper, the causes of equal-sound-path pseudo-images in the imaging area are modeled and analyzed. For the detection requirements of near-field defects in the R region of rib-stiffened wall panels, a dynamic pseudo-image removal scheme based on the effective equal-sound-path threshold is proposed by combining it with the simplest full-focusing algorithm, and the effectiveness of the algorithm is verified through comparative simulation imaging of near-field defects.

## 5. Ultrasonic Imaging Framework Design and System Setup

### 5.1. Setup of Ultrasonic Detection and Imaging System

In order to verify the imaging quality and performance of the minimum-variance beamforming (MVB) algorithm aimed at improving noise artifact imaging in ultrasonic phased-array imaging systems, an ultrasonic detection and imaging system was constructed and actual experiments were conducted. The experimental system, as shown in Figure 24, mainly consists of a phased-array excitation and receiving device, a 64-element ultrasonic phased-array probe, a computer host, and the test piece. The phased-array excitation and receiving device controls the phased-array element transmit–receive mode and collects and records ultrasonic echo signal data for upper-level transmission. The computer host is used to post-process the ultrasonic data for imaging, optimize the relevant algorithm, and output the imaging characterization results. In the construction process of the ultrasonic imaging experimental platform, the acoustic-field environment parameters of the detection system and the ultrasonic phased-array parameters are the same as the initialization settings of the simulation.

At the same time, in order to evaluate and analyze the effectiveness of the designed algorithm in practice, a batch of R-area aluminum alloy specimens were designed and customized. The shape, size, and material parameters of the specimens are the same as the outer thin-walled R-area structure that needs to be detected in actual engineering projects. By processing some of the R-area samples with common defects, the structural damage that may occur in the wing wall during actual work is simulated, and then the simulated defects are detected and characterized by ultrasonic imaging. The R-area structural customized specimens and defect processing situation based on proportional design are shown in Figure 25.

### 5.2. Defective Component Imaging Characterization

To simulate the actual scanning and imaging mode, the ultrasonic detection scheme for wall panel R-zone defects was followed, and the R-zone structural samples were fixed with a positive fixture. After treatment with ultrasonic phased-array coupling agent, they were tightly attached to the backs of the samples and scanned frame by frame. Utilizing the simplest total-focusing method algorithm, combined with adaptive beamforming and equiphase pseudo-image removal methods, the acquired phased-array echoes were post-processed to achieve imaging characterization of both defective and non-defective samples, and the characterization results were compared and analyzed. The imaging range of the *x*-axis was −20 mm to 20 mm, and the imaging range of the *z*-axis was 0 mm to 50 mm. The imaging results are shown in Figure 26.

From the imaging results, it can be seen that the noise and artifact components within the imaging area of the R region are well controlled, and the imaging characterization is clear and visible. Among them, the imaging results of the non-defective part of the sample can qualitatively characterize the through-hole structure of the wall panel sample, the bottom echo is arranged neatly, and the overall imaging is complete. At the same time, the imaging results of the non-defective part of the sample can also describe the position and shape of small defects. Due to the fact that some defects are located above the sample through-hole structure and the bottom plane, there is occlusion of the sample’s own structural morphology, which causes the ultrasonic signal to be unable to reach below the defect, resulting in partial imaging loss. However, through comparison with the imaging results of non-defective parts, it is still possible to achieve judgment of the detected defects. In summary, the proposed imaging algorithm can accurately reconstruct and clearly display the defect situation in the R-region structure of the wing wall panel, and the actual scanning and imaging time is stable at around 0.5 s/frame, which can achieve preliminary real-time detection and imaging of defects in the R-region structure.

In order to further quantitatively verify and analyze the performance of the imaging detection mode in the ultrasound scanning process, ultrasound imaging of the detection standard part was performed to obtain quantitative indicators describing the imaging resolution and sensitivity index requirements of the engineering project.

The ultrasonic wave velocity of the inspection standard specimen is 6300 m/s, and the structural dimensions are shown in Figure 27. The overall size of the standard specimen is 60 mm × 10 mm × 30 mm, and the diameters of the longitudinal sections of the through-hole structures from left to right are 0.1 mm, 0.2 mm, 0.3 mm, 0.4 mm, 0.5 mm, 1.5 mm, and 3.0 mm. The transverse spacing of the through-hole center is 3 mm, and the center depth of the through-hole is 2.5 mm. Since the energy of the ultrasonic signal attenuates with the increase in detection depth, theoretically, the detection resolution and sensitivity performance will decrease as the detection depth increases. In order to verify the imaging performance indicators of the imaging algorithm within the imaging range of 30 mm × 30 mm, the through-hole structure of the standard specimen is detected away from the surface of the ultrasonic phased-array element. At this time, the center depth of the through-hole is 27.5 mm, and the depth difference between through-holes of different sizes can be ignored. According to the control process and imaging mode of ultrasonic detection, the detection imaging algorithm is applied to characterize and analyze the standard specimen, and the imaging results are shown in Figure 28.

It can be seen from the figure that with the increase in the size of the holes, the imaging characterization quality of the algorithm gradually deteriorates, but it can still recognize the existence of holes, and at the same time, the bottom echo between the holes of the image is suppressed to a certain extent, avoiding its influence on defect imaging. For the overall imaging results, the through-hole structure at the bottom of the standard parts can be detected and characterized by the ultrasonic imaging algorithm, the detection sensitivity reaches 0.1 mm, the imaging results can distinguish and characterize the 2 through-hole structures on the right side that are the closest, and the lateral resolution reaches 0.75 mm, which meets the engineering index requirements of defect detection in the R area of the ribbed wall plate.

Research manuscripts reporting large datasets that are deposited in a publicly available database should specify where the data have been deposited and provide the relevant accession numbers. If the accession numbers have not yet been obtained at the time of submission, please state that they will be provided during review. They must be provided prior to publication.

Interventionary studies involving animals or humans and other studies that require ethical approval must list the authority that provided approval and the corresponding ethical approval code.

## 6. Conclusions

This paper mainly focuses on ultrasonic rapid detection and imaging algorithms for the defects in wing stringer wall panels, achieving a lateral resolution of 1 mm and a detection sensitivity of 0.5 mm within a single frame of 30 mm × 30 mm detection area. Compared with traditional single-element ultrasonic detection technology, ultrasonic phased-array detection has better defect echo quality, higher image signal-to-noise ratio, wider scanning range, and better adaptability to complex detection scenarios. Ultrasonic signal post-processing can flexibly process the collected ultrasonic wave signals to obtain better defect reconstruction images. Applying data post-processing technology to ultrasonic phased-array inspection systems can fully exploit the advantages of phased-array technology and effectively improve the imaging quality and defect detection rate. Therefore, based on the ultrasonic post-processing algorithm, the ultrasonic phased-array detection and imaging system has become a feasible research solution for the rapid detection and imaging of defects in wing stringer wall panels. However, due to the complex structure of the R area of the stringer wall panel and the rapid scanning requirements for defect detection, determining how to quickly and stably obtain the characteristic information of defects from ultrasonic echo data, avoid missed detections and false detections, and ensure the accuracy and robustness of the detection results, remains a technical problem that urgently needs to be solved.

## Figures and Tables

**Figure 1 sensors-23-05788-f001:**
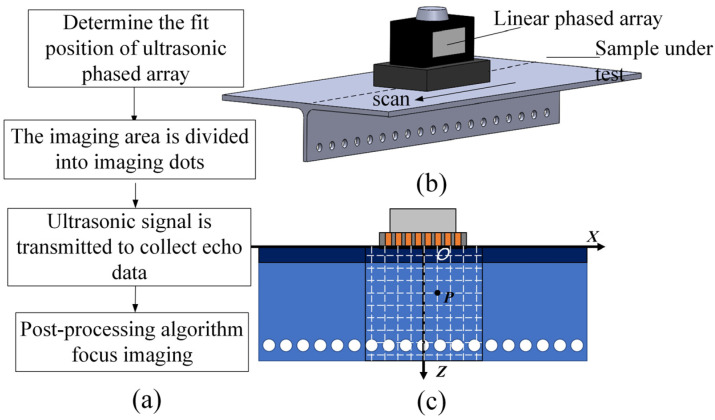
Schematic diagram of ultrasonic detection imaging for R-region defects. (**a**) Flowchart of phased-array detection. (**b**) Schematic diagram of phased-array scanning. (**c**) Coordinate system of detection front.

**Figure 2 sensors-23-05788-f002:**
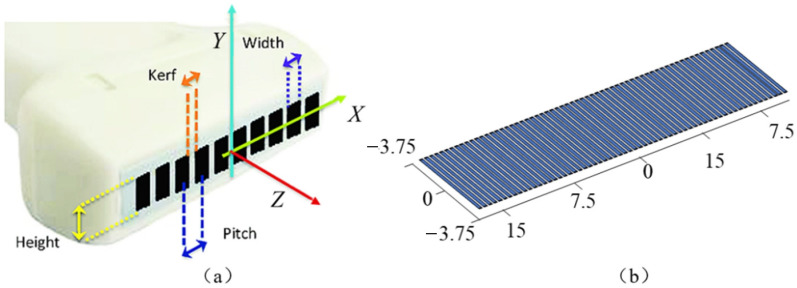
Simulation model of ultrasonic phased-array probe. (**a**) Phased-array coordinate distribution. (**b**) Element size model.

**Figure 3 sensors-23-05788-f003:**
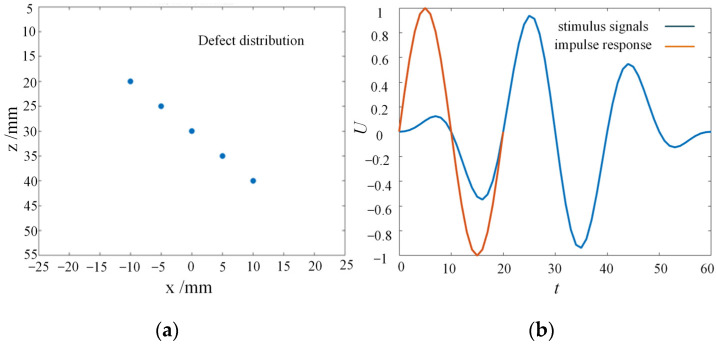
Detection waveforms and defect settings. (**a**) Excitation signal and impulse response waveform. (**b**) Defect distribution in the imaging area.

**Figure 4 sensors-23-05788-f004:**
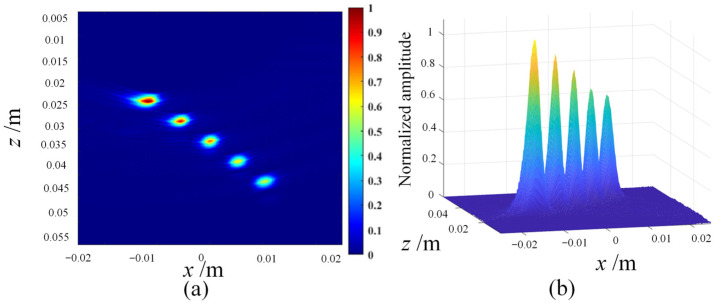
Simulation results of CPWC algorithm. (**a**) Grayscale amplitude of the imaging. (**b**) Normalized synthetic-aperture distribution.

**Figure 5 sensors-23-05788-f005:**
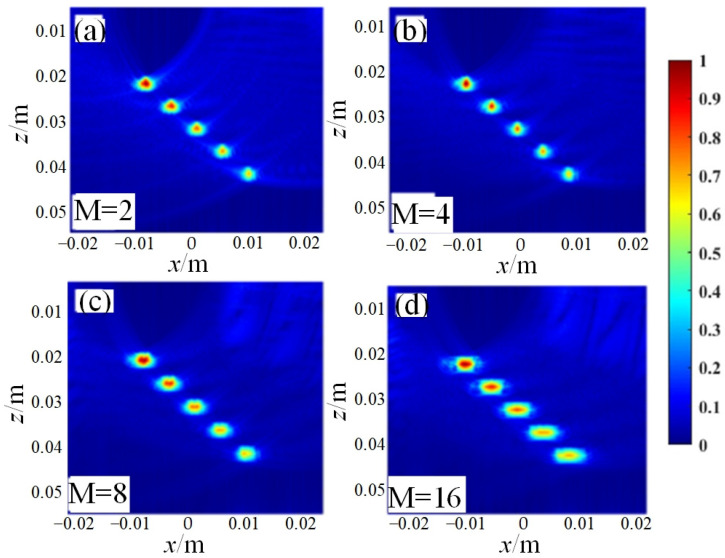
Simulation results of the MSAF algorithm for imaging area with different sub-aperture sizes. (**a**) M = 2, (**b**) M = 4, (**c**) M = 8, (**d**) M = 16.

**Figure 6 sensors-23-05788-f006:**
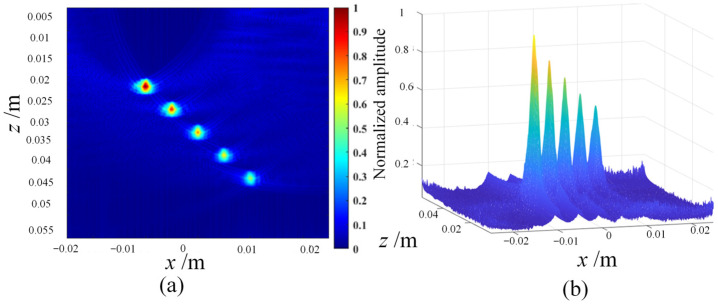
Simulation imaging results of the MSAF algorithm. (**a**) Imaging grayscale amplitude. (**b**) Normalized synthetic-aperture distribution.

**Figure 7 sensors-23-05788-f007:**
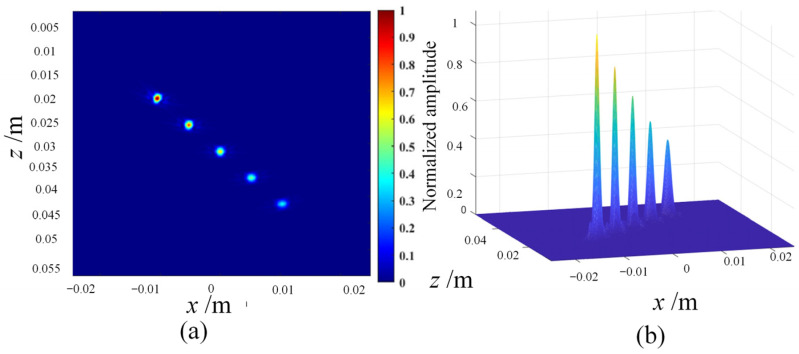
Simulation imaging results of the FMC-TFM algorithm. (**a**) Imaging grayscale amplitude. (**b**) Normalized synthetic-aperture distribution.

**Figure 8 sensors-23-05788-f008:**
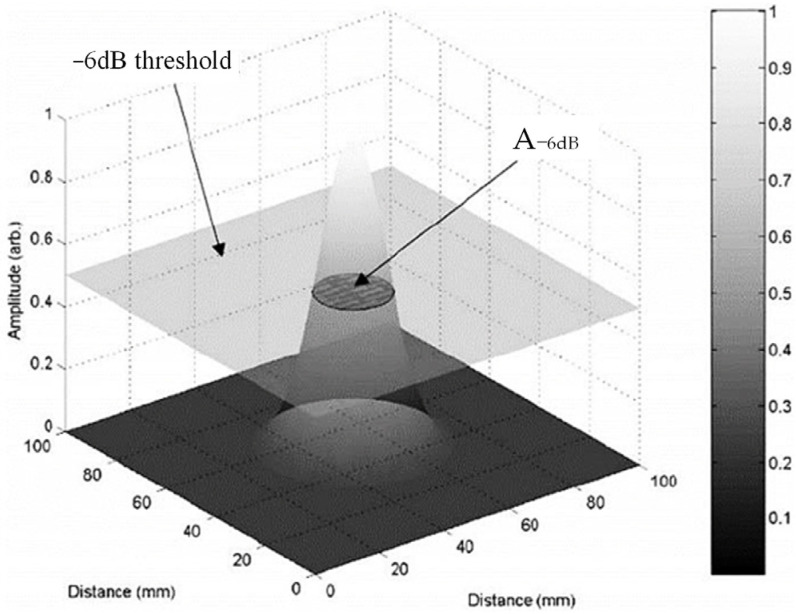
API calculation schematic.

**Figure 9 sensors-23-05788-f009:**
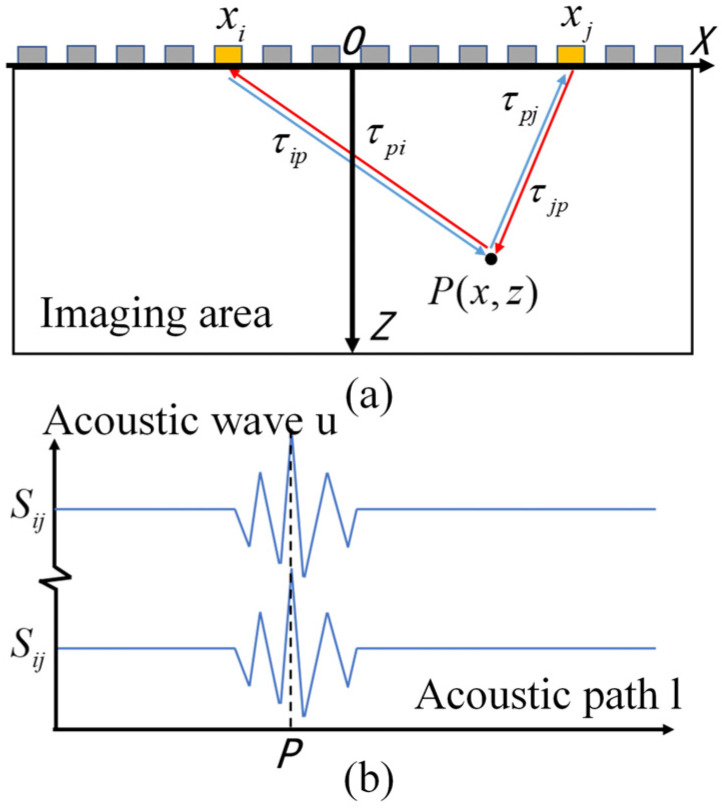
Schematic diagram of full-matrix data reciprocity. (**a**) Symmetrical channel propagation path. (**b**) Symmetrical channel A-scan signal.

**Figure 10 sensors-23-05788-f010:**
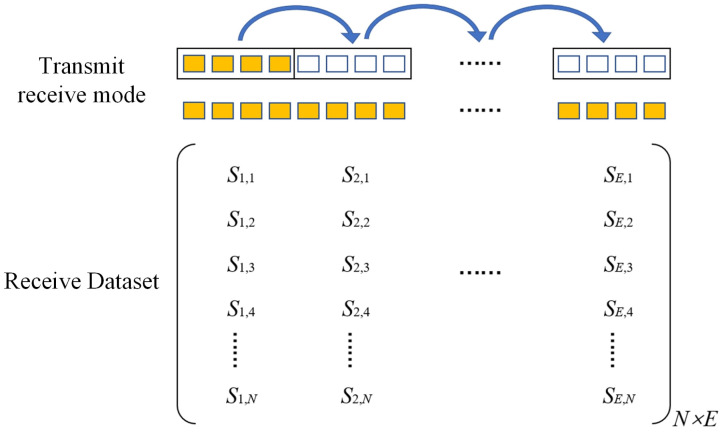
Schematic diagram of element transmit/receive control for STA imaging (with 4 sub-aperture elements).

**Figure 11 sensors-23-05788-f011:**
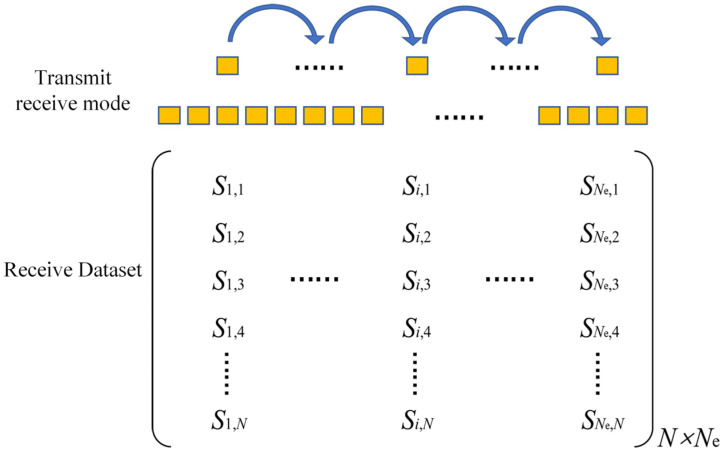
Schematic diagram of element transmit/receive control for SMC-TFM imaging.

**Figure 12 sensors-23-05788-f012:**
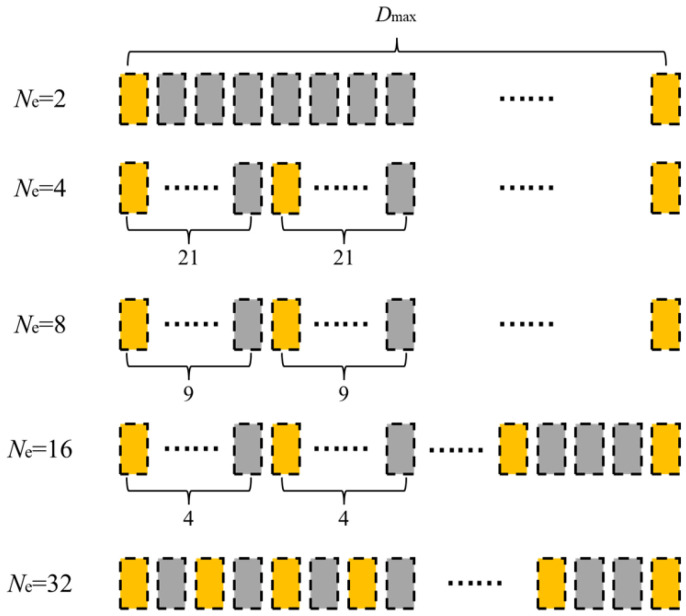
Schematic diagram of element sparsity pattern for SMC-TFM imaging.

**Figure 13 sensors-23-05788-f013:**
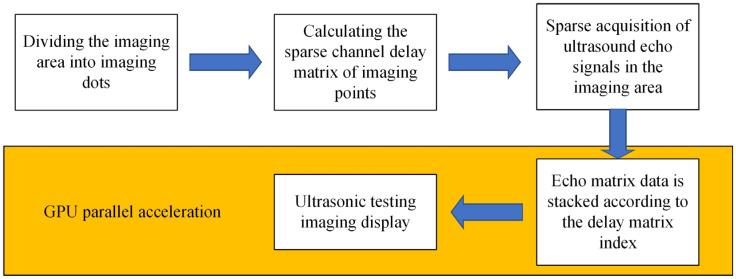
Flowchart of the simplest total-focusing method algorithm.

**Figure 14 sensors-23-05788-f014:**
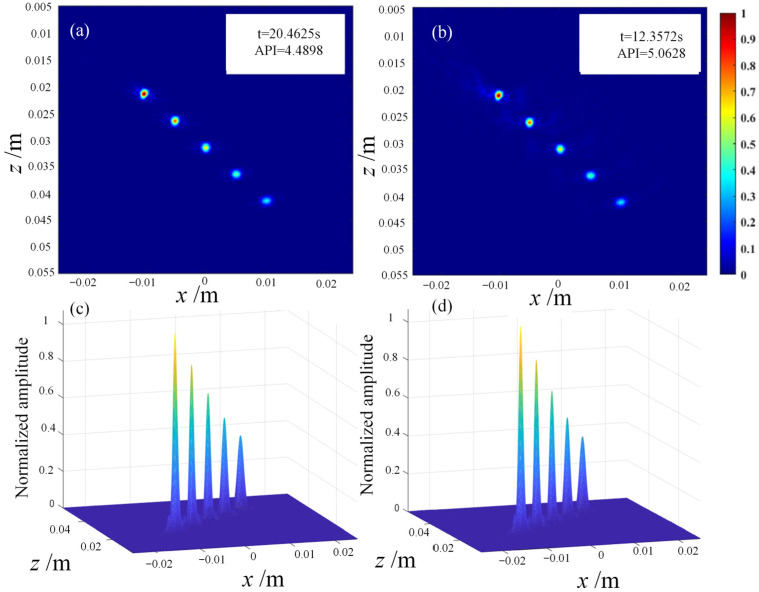
Simulation imaging comparison between full-matrix algorithm and 1/2 matrix algorithm. (**a**) Grayscale imaging of full-matrix algorithm. (**b**) Grayscale imaging of 1/2 matrix algorithm. (**c**) Synthetic beam of full-matrix algorithm. (**d**) Synthetic beam of 1/2 matrix algorithm.

**Figure 15 sensors-23-05788-f015:**
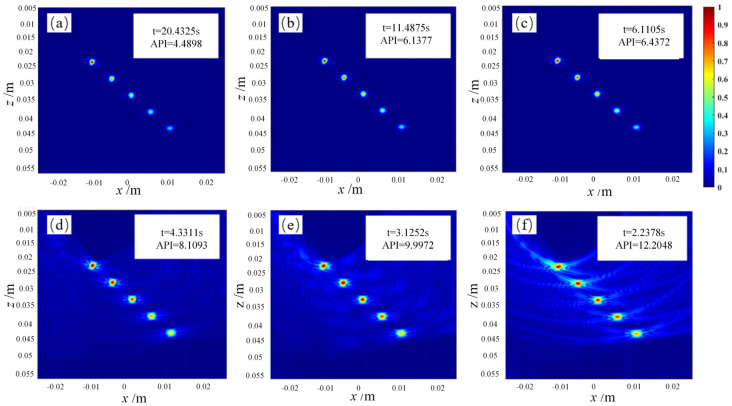
Simulation imaging results for different sub-aperture sizes in STA imaging. (**a**) M = 1, (**b**) M = 2, (**c**) M = 4, (**d**) M = 8, (**e**) M = 16, (**f**) M = 32.

**Figure 16 sensors-23-05788-f016:**
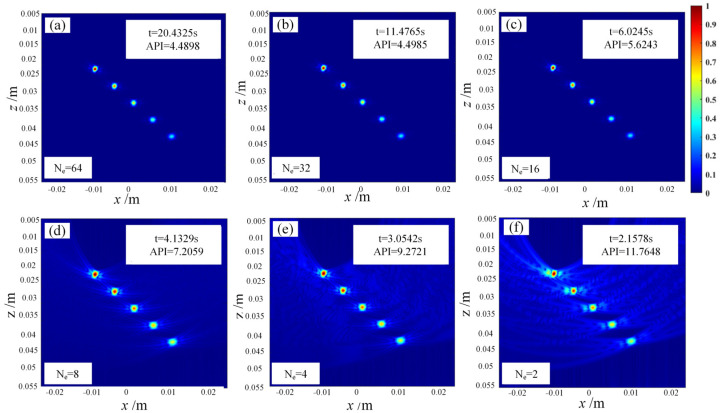
Simulation imaging results for sparse total-focusing method. (**a**) Ne = 64, (**b**) Ne = 32, (**c**) Ne = 16, (**d**) Ne = 8, (**e**) Ne = 4, (**f**) Ne = 2.

**Figure 17 sensors-23-05788-f017:**
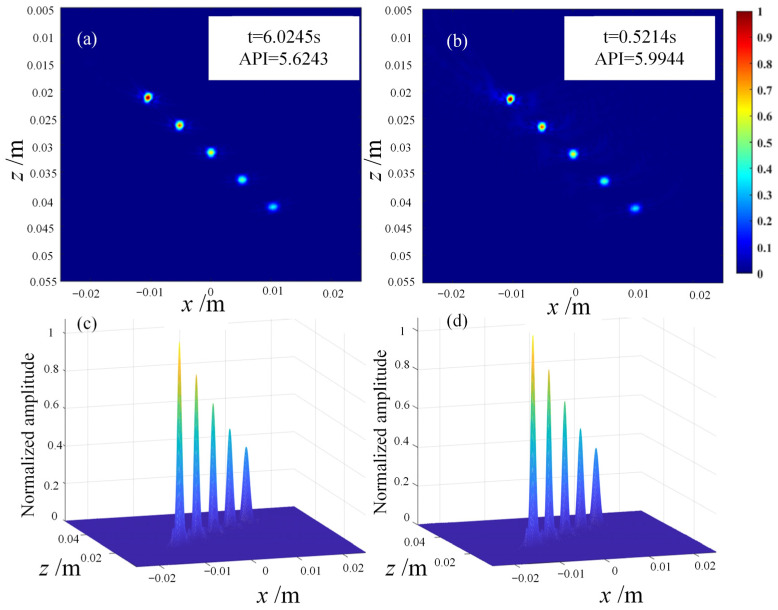
Simulation imaging comparison between SMC-TFM algorithm and STFM algorithm (Ne = 16). (**a**) Grayscale imaging of SMC-TFM algorithm. (**b**) Grayscale imaging of STFM algorithm. (**c**) Synthetic beam of SMC-TFM algorithm. (**d**) Synthetic beam of STFM algorithm.

**Figure 18 sensors-23-05788-f018:**
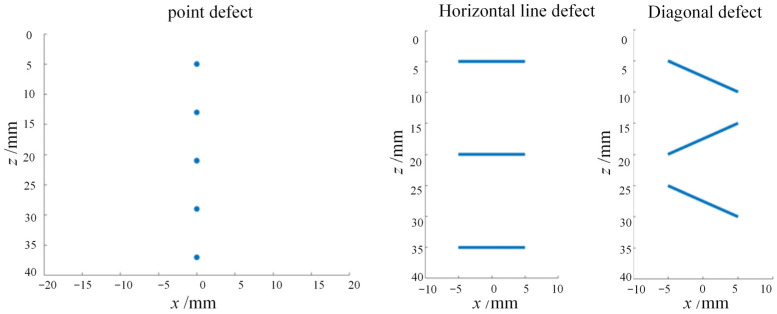
Simulation defect distribution settings.

**Figure 19 sensors-23-05788-f019:**
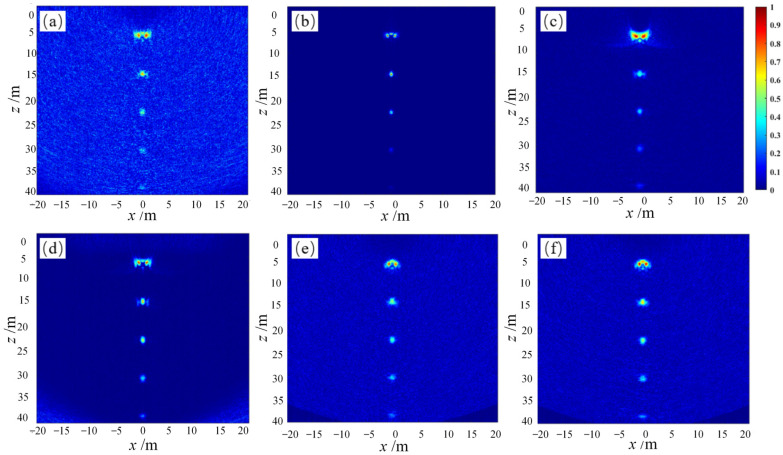
Simulation imaging of point targets using adaptive beamforming algorithms. (**a**) DAS algorithm imaging, (**b**) CF algorithm imaging, (**c**) GCF algorithm imaging, (**d**) SCF algorithm imaging, (**e**) MV algorithm imaging, (**f**) ESBMV algorithm imaging.

**Figure 20 sensors-23-05788-f020:**
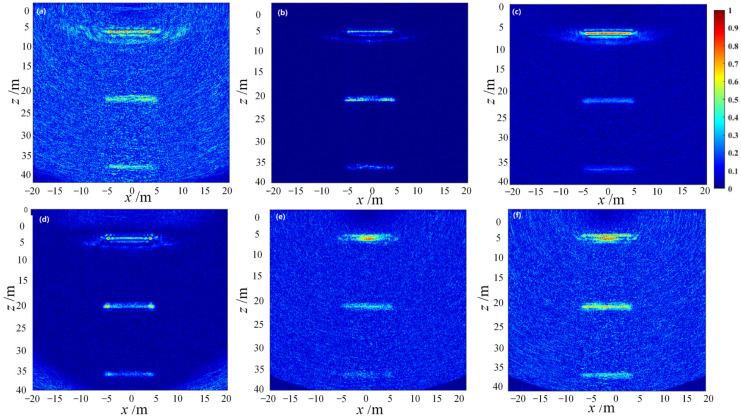
Simulation imaging of horizontal line targets using adaptive beamforming algorithms. (**a**) DAS algorithm imaging, (**b**) CF algorithm imaging, (**c**) GCF algorithm imaging, (**d**) SCF algorithm imaging, (**e**) MV algorithm imaging, (**f**) ESBMV algorithm imaging.

**Figure 21 sensors-23-05788-f021:**
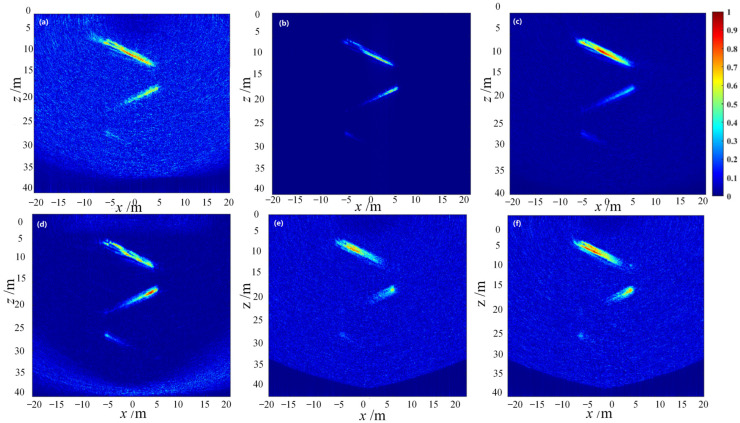
Simulation imaging of inclined line targets using adaptive beamforming algorithms. (**a**) DAS algorithm imaging. (**b**) CF algorithm imaging. (**c**) GCF algorithm imaging. (**d**) SCF algorithm imaging. (**e**) MV algorithm imaging. (**f**) ESBMV algorithm imaging.

**Figure 22 sensors-23-05788-f022:**
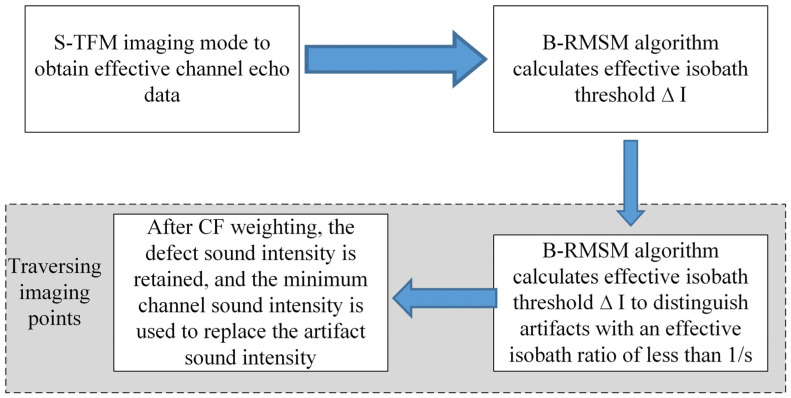
Flowchart of the equivalent-sound-path error elimination algorithm.

**Figure 23 sensors-23-05788-f023:**
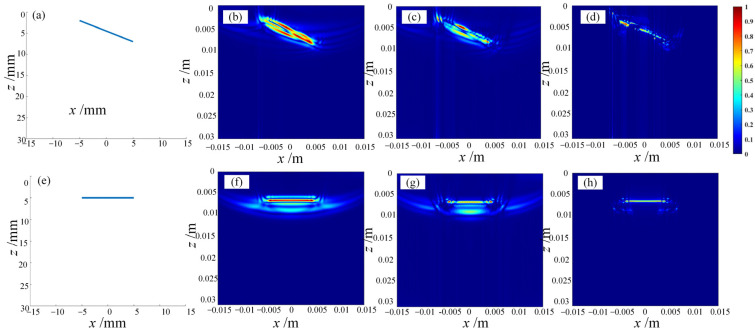
Simulation imaging comparison of the equivalent-sound-path pseudo-image removal algorithm. (**a**,**e**) show the distributed line defects in the near-field region, and (**b**–**d**) are the imaging results of the DAS algorithm, the SCF algorithm, and the EAEM algorithm for oblique line defects, respectively. (**f**–**h**) are the imaging results of the DAS algorithm, the SCF algorithm, and the EAEM algorithm for oblique line defects, respectively.

**Figure 24 sensors-23-05788-f024:**
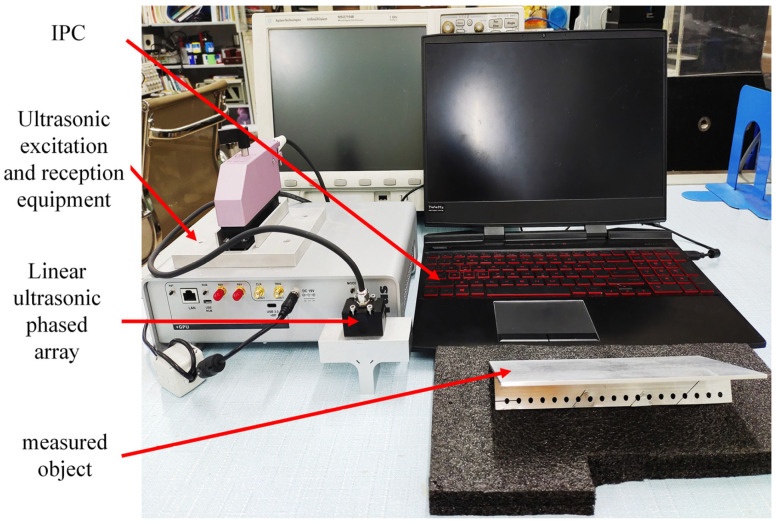
Experimental setup of ultrasonic detection and imaging system.

**Figure 25 sensors-23-05788-f025:**
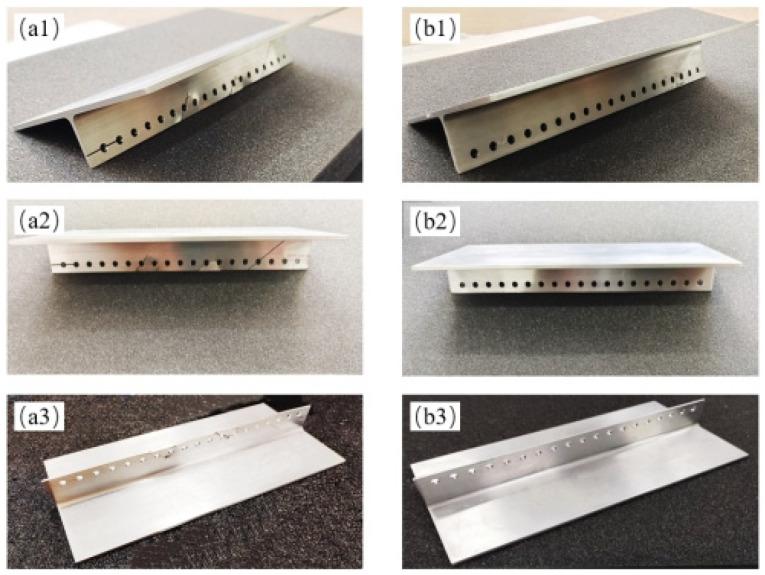
Customized wing wall R-area structural specimens. (**a1**–**a3**) Defective specimens, (**b1**–**b3**); Non-defective specimens.

**Figure 26 sensors-23-05788-f026:**
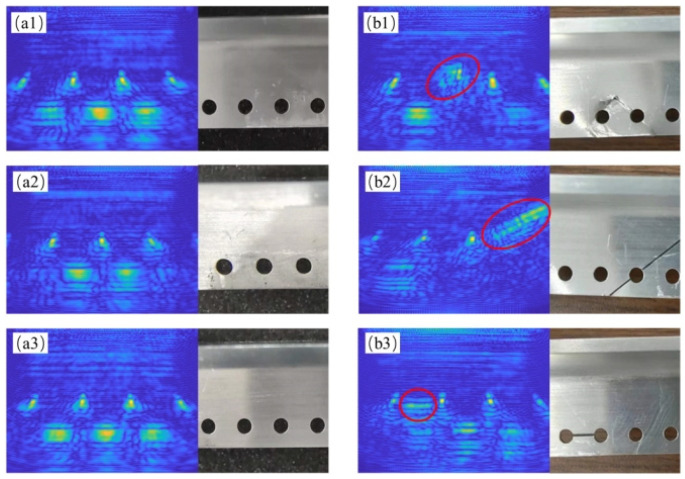
Imaging results and corresponding physical structures of defect and non-defect samples. (**a1**–**a3**) Non-defect sample imaging. (**b1**–**b3**) Defect sample imaging.

**Figure 27 sensors-23-05788-f027:**
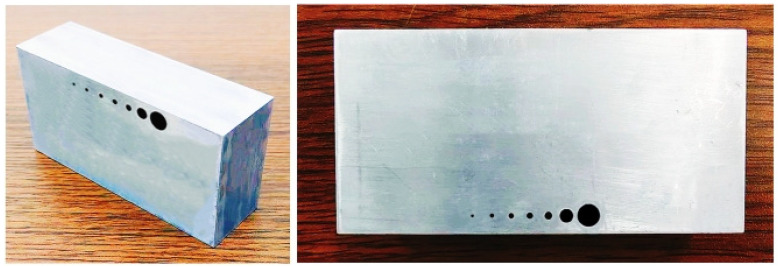
Test standard component.

**Figure 28 sensors-23-05788-f028:**
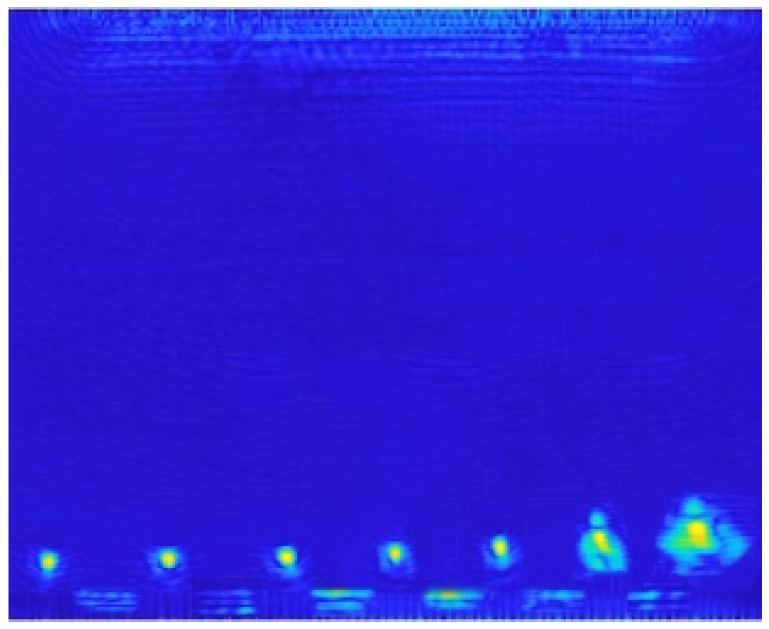
Ultrasonic imaging results of the standard specimen.

**Table 1 sensors-23-05788-t001:** Comparison of Mainstream Non-Destructive Testing Methods.

Non-Destructive Testing Method	Principle	Advantages	Disadvantages	Defect Features
Magnetic Particle Testing	Magnetic particles visualized by the leakage magnetic field at defect areas	Intuitive display, high sensitivity, simple process, low cost	Limited to ferromagnetic materials, difficult to achieve automation	Surface and near-surface defects
Eddy Current Testing	Detection of eddy currents generated by alternating magnetic fields	Non-contact inspection, fast detection, high sensitivity	Difficult to quantitatively characterize, low detection depth, low accuracy	Surface defects
Liquid Penetrant Testing	Amplification of defects through penetration of liquid penetrant	Not limited by material type and external shape constraints, high sensitivity	Slow detection speed, low robustness, difficult to automate	Surface defects
X-ray Testing	Imaging based on sample’s absorption and scattering of X-rays	Strong penetration, not affected by external shape and structure, high efficiency	Harmful radiation effects on inspection personnel	Surface and internal defects

**Table 2 sensors-23-05788-t002:** Parameters for simulation experiment.

Parameter Type	Parameter	Setting
Acoustic Environment Parameter	Sound Velocity	6300 m/s
	Signal Sampling Rate	100 MHZ
Phased-Array Ultrasonic Parameter	Number of Phased-Array Elements	64
	Element Spacing	0.5 mm
	Element Length	10 mm
	Element Gap	0.1 mm
	Center Frequency	5 MHZ

**Table 3 sensors-23-05788-t003:** API index calculation results.

Imaging Algorithm	API Value
CPWC (M = 17)	26.0346
CPWC (M = 21)	29.1079
MSAF (M = 2)	12.3590
MSAF (M = 4)	8.7711
MSAF (M = 8)	16.6226
MSAF (M = 16)	23.7635
FMC-TFM	4.4898

## Data Availability

Data is unavailable due to privacy.

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
