# Peer review of "Defect Detection Algorithm for Wing Skin with Stiffener Based on Phased-Array Ultrasonic Imaging"

_sensors, 2023, doi:10.3390/s23135788_

Round 1

Reviewer 1 Report

This paper presents a defect detection algorithm based on phased array ultrasonic imaging for wing skin with stiffener. Although the study is interesting, the presentation needs substantial improvement. Some of the plots for results are of poor quality.

It is mentioned mathematical models are developed and simulation results based these models are produced. Some details about the mathematical models should be given.

Simulation is mentioned a few times when presenting the results. Are the results in this paper based on simulation? Are there any experimental results? Some clarification should be given.

In Figure 2-3, there are no (a) and (b). In the right half, nothing is plotted and the axes do not have labels. Also figures should be numbered based on sections and should be just Figure 1, Figure 2, etc.

Figures 2-3, 2-4, 2-6, and 2-7 are of poor quality.

Figure 3-1 is poor and unclear. What are you trying to show here and why many parts are empty?

Figure 3-3 is poor and it looks like something is missing.

In Figure 3-6, 3 of the 4 subplots are empty.

In Figure 3-8, 5 of the 6 subplots are empty.

In Figure 3-9, 3 of the 4 subplots are empty.

In Figure 4-6, 7 of the 8 subplots are empty.

Figure 5.1 seems having some parts missing.

English is overall ok.

Author Response

Please refer to the document “Response1.pdf" for specific review comments and responses

Reviewer 2 Report

In the present study is proposed a phased array measurement and imaging scheme for detecting defects in reinforced wall panels in the R region and the study of the post-processing imaging algorithms for ultrasonic detection

Although the subject is interesting, some parts of the text must be significantly improved before publication.

General comment:

Is the manuscript provided most of the figures have problems and it is not possible to visualize the intended information.

Examples are figures: 2-1 b), 2-3 a) b), 2-4, 2-6, 2-7, 3-1, 3-3, 3-5, 3-8, 3-9, 4-3, 4-4, 4-6 and 5.1. Without them being correct it is impossible to understand much of the information in the text.

Figures correction is mandatory for the present paper meets the requirements for the published

Detailed comments about the manuscript:

1. Page 2, line 93

Must be introduced more details about the method used to evaluate the images.

2. Page 3, line 100

What was the used software? Details about the simulation must be provided.

3. Page 4, line 136

The authors say: “The following are the simulation results of three post-processing algorithms, namely coherent plane-wave compounding imaging, multi-element synthetic aperture imaging, and full-matrix/full-focus imaging”. The number of the figures should be mentioned for each method.

4. Page 4, line 140

The author speaks about CPWC. Acronymous must be previous defined

5. Page 4, line 143

The author speaks about MSAF. Acronymous must be previous defined.

6. Page 6, line 158

In text: “…ultrasound wavelength square λ2…”, two should be the exponent.

7. Page 6, line 167

Previous the full-matrix full-focus imaging was defined as FMC-TFM and now FMFF. Coherence should be maintained.

The same comment to multi-element synthetic aperture (MESA), previously called MSAF, and hybrid plane wave (HPW), previously called CPWC.

8. Page 11, line 293

The authors say: “…the DAS, MV, ESBMV, CF, 293 GCF, and SCF beamforming algorithms…”. Must be mentioned the names that originated these acronymous.

9. Page 16, line 415

The authors say: “Through the above simulation experiments…”, is not possible to achieve any conclusion due to lack of information in the figures.

10. Page 17, line 448

Details about the defects (dimensions, orientation, format...) should be provided.

11. Page 18, line 452

In the legend of Figure 5: “(a1)(a2) Defective specimens, (b1)(b2) 452 Non-defective specimens.”, what about samples a3 and b3?

12. Page 18, line 467

Now "a " are non-defect samples and "b" are defect samples? It was the contrary previously.

13. Page 18, line 506

What are the conclusions obtained from figure 5.5?

Author Response

Please refer to the document "Response2.pdf" for specific review comments and responses

Reviewer 3 Report

This manuscript demonstrates the real-time detection methodology for defect detection in wing stringer wall panels based on Phased Array Ultrasonic Imaging. I am not able to make precise comments based on current version of manuscript because most of the figures are broken and with low resolution so that the data/results are blind to me. Authors need to re-submit the manuscript with complete, high-quality/resolution figures. Apart from the figure issue, there are questions/comments listed below:

1. In line 231-242 (last paragraph of section #3), authors mentioned that the scanning frequency got improved by introducing a de-symmetry redundant imaging mode. And the result is that speed is successfully being improved and meet the scanning requirements. Can authors quantify the speed improvement and how much speed gap between actual vs. requirement is before this implementation?

2. Some format misalignments: 1) the index of equations in page 15 are off format; 2) Figure 5-2 (written as Figure 5) caption misses (a3) and (b3)

Author Response

Please refer to the document "Response3.pdf" for specific review comments and responses

Reviewer 4 Report

This paper is devoted to the defect detection algorithm based on phased array ultrasonic imaging for wing skin with stiffener. To address noise artifacts, an adaptive beamforming method was proposed. In my opinion, the paper could be published, but after a revision. Before considering its publication, authors should address a few issues.

1.       What do “detection sensitivity of 1mm and a resolution of 0.5mm” mean? Could you describe it in more detailed way?

2.       I could not find in text where the authors describe the results they declared in the abstract and conclusion: "achieving a lateral resolution of 1mm and a detection sensitivity of 0.5mm within a single frame of 30mm × 30mm detection area". It should be pointed out more clearly.

3.       All of the figures have really poor quality. At least that is what I see when downloading the PDF file of the manuscript.

4.       figure 2-1 has poor quality and have to be re-designed. The sub-figure 2-1 (c) is misleading. If it is a coordinate system, then where are axes? It has to be done clearer.

5.       figure 2-2, 2-3 - again the quality of charts is poor

6.       figure 2-3, 4-3 - there are no labels "a", "b" on the figure

7.       figures 2-3, 2-4, 2-6, 2-7, 3-6, 3-8, 3-9, 4-3, 4-4, 4-6, 5-1 seem to be truncated or not loaded correctly. If the images are supposed to be like that, please clearly state it in the text. If not, please provide the correct pictures.

8.       figures 2-4, 2-5, 2-6, 2-7, 4-1 are not referenced in the text

9.       it is absolutely unclear what is presented on the figure 3-1

10.   please align the numbers of the formulas' 1,2,3,4 (the brackets and the number should be one-lined)

11.   line 40-44 - the authors define what is non-destructive testing, provide examples of testing, i.e., by utilizing changes in thermal, acoustic, optical, electrical, magnetic reactions, but then talking about the advantages of ultrasonic technique only. The same is for adaptive beam forming algorithm. It would be better to briefly describe the pros and cons of the other techniques they just said about. What can authors say about optical and magnetic techniques listed below?

·         Y.-K. Zhu, G.-Y. Tian, R.-S. Lu, H. Zhang, A Review of Optical NDT Technologies, Sensors 11, pp. 7773-7798, 2011. https://doi.org/10.3390/s110807773

·         C. Fan, M. Caleap, M. Pan, B. Drinkwater, A comparison between ultrasonic array beamforming and super resolution imaging algorithms for non-destructive evaluation, Ultrasonics 54(7), pp. 1842-1850, 2014. https://doi.org/10.1016/j.ultras.2013.12.012 .

·         I. Galaktionov, A. Nikitin, J. Sheldakova, A. Kudryashov, "B-spline approximation of a wavefront measured by Shack-Hartmann sensor", Proc. SPIE 11818, pp.118180N, 2021. https://doi.org/10.1117/12.2598249

·         H. Zhang, R. Yang, Y. He, A. Foudazi, L. Cheng, G. Tian, A Review of Microwave Thermography Nondestructive Testing and Evaluation, Sensors, 17, pp. 1123, 2017. https://doi.org/10.3390/s17051123

·         B. Hu, R. Yu, H. Zou, Magnetic non-destructive testing method for thin-plate aluminum alloys, NDT & E International, 47, pp. 66-69, 2012. https://doi.org/10.1016/j.ndteint.2011.12.007

·         C. Fan, B. Drinkwater, Comparison between beamforming and super resolution imaging algorithms for non-destructive evaluation. AIP Conference Proceedings 18, 1581, pp. 171–178, 2014. https://doi.org/10.1063/1.4864817

·         J. Sheldakova, V. Toporovsky, I. Galaktionov, A. Nikitin, A. Rukosuev, V. Samarkin, A. Kudryashov, Flat-top beam formation with miniature bimorph deformable mirror, Proc. SPIE 11486, pp. 114860E, 2020. https://doi.org/10.1117/12.2569387

·         R. Yu, B. Hu, H. Zou, W. Xiao, Q. Cheng, W. Xu, J. Xin, Micromagnetic technology for detection of carbon impurity in crystalline silicon, NDT & E International 62, pp. 1-5, 2014. https://doi.org/10.1016/j.ndteint.2013.10.003 .

In my opinion, putting a small paragraph that briefly describes the techniques in works listed above will define the place of this particular work amongst other researches.

12.   There are a few grammatical errors that have to be addressed. Below just a few examples.

-          line 13 - "The algorithm selects the full-matrix full-focusing algorithm" - the sentence should be corrected

-          line 55-58 - the sentence has to be re-written, there are too much word duplications.

-          line 78-79 - "evaluates the best focusing imaging method" - I believe there should either focusing or imaging but not both of them

-          line 123 - there are typos in the imaging ranges, i.e., "range of 5mm55mm"

Overall quality of english is good, though a few corrections have to be done to improve it

Author Response

Please refer to the document "Response4.pdf" for specific review comments and responses

Round 2

Reviewer 1 Report

The authors have adequately addressed my comments and the revised manuscript can be accepted.

Reviewer 2 Report

After the revision made by the authors the paper can be accepted in the current form.

Reviewer 3 Report

Authors added higher quality of figures which correlate more to the model and experiment designed in the manuscript. Moreover, my questions have been addressed and reflected in the manuscript. I would recommend to accept it in current form.